

# Integrating robotics into wildlife conservation: testing improvements to predator deterrents through movement

Stewart W. Breck[1], Jeffrey T. Schultz[1], David Prause[2], Cameron Krebs[3], Anthony J. Giordano[2,4] and Byron Boots[5]

[1] USDA-WS-National Wildlife Research Center, Fort Collins, Colorado, United States
[2] Center for Human-Carnivore Coexistence, Colorado State University, Fort Collins, Colorado, United States
[3] Krebs Livestock, Ione, Oregon, United States
[4] S.P.E.C.I.E.S.—The Society for the Preservation of Endangered Carnivores and Their International Ecological Study, Ventura, California, United States
[5] Paul G. Allen School of Computer Science and Engineering, University of Washington, Seattle, Washington, United States

## ABSTRACT

**Background:** Agricultural and pastoral landscapes can provide important habitat for wildlife conservation, but sharing these landscapes with wildlife can create conflict that is costly and requires managing. Livestock predation is a good example of the challenges involving coexistence with wildlife across shared landscapes. Integrating new technology into agricultural practices could help minimize human-wildlife conflict. In this study, we used concepts from the fields of robotics (*i.e.*, automated movement and adaptiveness) and agricultural practices (*i.e.*, managing livestock risk to predation) to explore how integration of these concepts could aid the development of more effective predator deterrents.

**Methods:** We used a colony of captive coyotes as a model system, and simulated predation events with meat baits inside and outside of protected zones. Inside the protected zones we used a remote-controlled vehicle with a state-of-the art, commercially available predator deterrent (*i.e.*, Foxlight) mounted on the top and used this to test three treatments: (1) light only (*i.e.*, without movement or adaptiveness), (2) predetermined movement (*i.e.*, with movement and without adaptiveness), and (3) adaptive movement (*i.e.*, with both movement and adaptiveness). We measured the time it took for coyotes to eat the baits and analyzed the data with a time-to-event survival strategy.

**Results:** Survival of baits was consistently higher inside the protected zone, and the three movement treatments incrementally increased survival time over baseline except for the light only treatment in the nonprotected zone. Incorporating predetermined movement essentially doubled the efficacy of the light only treatment both inside and outside the protected zone. Incorporating adaptive movement exponentially increased survival time both inside and outside the protected zone. Our findings provide compelling evidence that incorporating existing robotics capabilities (predetermined and adaptive movement) could greatly enhance protection of agricultural resources and aid in the development of nonlethal tools for managing wildlife. Our findings also demonstrate the importance of marrying

Corresponding author
Stewart W. Breck,
stewart.w.breck@usda.gov

agricultural practices (*e.g.*, spatial management of livestock at night) with new technology to improve the efficacy of wildlife deterrents.

## INTRODUCTION

The impacts of wildlife on agriculture are complex, global issues often with adverse outcomes for both wildlife and agricultural producers (*Nyhus, 2016*). The financial burden of wildlife preying on farm animals, eating crops, and damaging farmland is estimated to be in the billions of dollars annually in the U.S. alone (*Conover & Conover, 2022*; *Reidinger, 2022*). Reducing this damage is a complex socioeconomic challenge because the public demands both low-cost food and minimal negative impacts to the environment, including native wildlife populations (*Seoraj-Pillai & Pillay, 2017*). The discovery of better methods to coexist with wildlife will likely benefit from a merger of enhanced technology with a better understanding of optimal agricultural practices (*Holloway, 2007*; *Maldonado et al., 2008*; *Nabokov et al., 2020*; *Paranjape et al., 2018*). As more technology becomes available (*e.g.*, developments in unmanned aerial systems and robotics) to potentially aid in wildlife management (*Egan et al., 2020*; *Ghobadpour et al., 2022*; *Hahn et al., 2017*; *Roshanianfard et al., 2020*), it is important to test where integration might be beneficial and demonstrate which advances are the most promising. In this study, our aim was to utilize a model predator system to test concepts made possible by integrating advances in robotics with established agricultural practices to determine the potential benefits of new technologies in reducing wildlife damage to agriculture, including predation by carnivores.

Predation of livestock by carnivores is an important issue in many parts of the world where concern for the ecological functioning and sustainability of carnivores overlaps with human livelihoods (*Baker et al., 2008*). For example, in the US, livestock predation results in roughly $300M annually in financial burdens for landowners (*NAHMS, 2015a*, *2015b*) and threatens the sustainability of small, rural ranching operations that have less capacity to absorb these burdens (*Muhly & Musiani, 2009*; *Ramler et al., 2014*; *Seoraj-Pillai & Pillay, 2017*; *Steele et al., 2013*). Given societal goals of conserving carnivore populations (*Mech, 1996*), there is growing demand for approaches that can reduce livestock losses and simultaneously maintain thriving agribusinesses and carnivore populations (*Venumiere-Lefebvre, Breck & Crooks, 2022*).

There are many and varied approaches to enhancing human-carnivore coexistence (*Smith et al., 2000a*, *2000b*). One approach is to focus on management of livestock with practices like utilizing livestock breeds that are aggressive toward predators, enhancing livestock health to reduce vulnerability, and managing the spatial distribution of livestock on a landscape (*Bruns, Waltert & Khorozyan, 2020*; *Muhly et al., 2010*). Of these, managing livestock dispersion to reduce predation is the most studied and commonly implemented management practice (*Ogada et al., 2003*; *Robel et al., 1981*). Examples of this type of

practice include the use of herding, and night pens (*Smith et al., 2000a*). Another approach is to utilize wildlife deterrents (*e.g.*, light or sound devices) that alter the appetitive behaviors of carnivores (*Shivik, Treves & Callahan, 2003*). This is an important class of tools because deterrents offer a lot of potential for incorporating new technology (*Miller et al., 2016*; *Naha et al., 2020*). For example, the technology integrated into drones has expanded rapidly in the past decade and is creating more opportunities in wildlife conservation (*Mo & Bonatakis, 2022*), agriculture (*Rejeb et al., 2022*) and managing human-wildlife conflict (*Hahn et al., 2017*; *Wandrie, Klug & Clark, 2019*). Importantly, the efficacy of wildlife deterrents also relates to how deterrents are integrated with livestock management (*Khorozyan & Waltert, 2019*; *Miller et al., 2016*), with greater success resulting from the combined use of livestock management and wildlife deterrents (*Lesilau et al., 2018*; *Stone et al., 2017*).

A variety of studies have tested the effectiveness of deterrents for reducing wildlife damage generally (*Gilsdorf, Hygnstrom & VerCauteren, 2003*; *VerCauteren et al., 2020*) and carnivore depredation specifically (*Khorozyan & Waltert, 2019*; *Miller et al., 2016*; *Naha et al., 2020*). Lessons from this and other work indicate that at least three concepts are important for making deterrents effective: (1) proximity, (2) unpredictability, and (3) adaptiveness. With respect to proximity, the closer a deterrent is to a target animal the greater the impact. This is one of the justifications for creating deterrents that are activated by individual wildlife. For example, an animal-activated deterrent for scaring deer was established using a monofilament fence around a particular crop. When the deer were in close proximity, they broke the monofilament line, activating the deterrent (*Beringer, VerCauteren & Millspaugh, 2003*). Secondly, a unpredictable deterrent will maintain its novelty longer than one that is predictable (*Shivik, 2006*). There are various ways of incorporating this concept into deterrents. Some examples of unpredictability in deterrents include using randomly flashing lights (*Linhart et al., 1992*), including different colored lights (*e.g.*, Foxlight-see below), and incorporating sound as well as light (*VerCauteren, Shivik & Lavelle, 2005*). Finally, deterrent adaptiveness considers the degree to which a deterrent can sense and react to the presence and behavior of wildlife (*Shivik, 2006*); deterrents with more adaptiveness are generally more effective. An example of a deterrent with greater adaptiveness is the radio-activated guard, which contains a VHF receiver that is triggered when animals with a radio-collar come within a certain distance of the device (*Breck et al., 2002*).

The emergence of autonomous vehicles and related technologies in agriculture (*Ghobadpour et al., 2022*; *Roshanianfard et al., 2020*) opens many new paths for improving deterrents. There are no existing examples of terrestrial based mobile deterrents and adding mobility would likely improve overall effectiveness by increasing performance in all three of the key areas: proximity, unpredictability, and adaptiveness. For example, GPS route following, RGB (red, green and blue) and thermal camera image recognition, and real-time AI image processing could lead to a deterrent capable of (1) moving across landscapes at set times and with set routes (*i.e.*, predetermined movement) and (2) a deterrent capable of moving in reaction to animals on the landscape (*i.e.*, adaptive movement). However, because these emerging technologies are relatively expensive,

gaining greater understanding about the potential benefits of increasing mobility would help understand whether investment in this area is justified.

In this study we combined the idea of enhancing deterrent effectiveness with the idea of spatial management of livestock to evaluate the importance of integrating these ideas. We used a colony of captive coyotes as a model system and simulated predation events using edible baits to perform our experiment. Coyotes are the most important predator of livestock in the U.S. (*Knowlton, Gese & Jaeger, 1999*; *Mitchell, Jaeger & Barrett, 2004*) and their range in North America has expanded tremendously in the past 50 years (*Hody & Kays, 2018*; *Poessel, Gese & Young, 2017*). Coyotes are intelligent, generalist omnivores (*Gese, Ruff & Crabtree, 1996*) and there has been extensive research on the development of new techniques for preventing coyote conflict (*Knowlton, Gese & Jaeger, 1999*; *Mitchell, Jaeger & Barrett, 2004*), including the use of deterrents (*Linhart et al., 1992*; *Windell et al., 2022*; *Young, Draper & Breck, 2019*). To our knowledge, no work has been done on coyotes that integrates the concepts of mobility for enhancing deterrent effectiveness. By utilizing captive coyotes, we were able to perform a tightly controlled experiment where we could test predetermined and adaptive movement of deterrents and evaluate whether further development was justified.

Our first objective was to evaluate the concept of mobility and how it could improve current state of the art wildlife deterrents. We tested this using a controlled experiment that protected baits from being eaten by coyotes through the implementation of three levels of a treatment: (1) light only, (2) predetermined movement (*i.e.*, movement at a predetermined time and in a set pattern, not in response to coyotes), and (3) adaptive movement (*i.e.*, movement in response to approaching coyotes). Our second objective was to simulate a landscape that varies in predation risk (*i.e.*, we varied bait proximity to the location of the deterrent) to test for an interactive effect between the movement treatments and predation risk. Though we used coyotes to carry out this experiment, we believe the concepts tested in this experiment potentially apply to a wide variety of wildlife.

## MATERIALS AND METHODS

This study was conducted at the USDA National Wildlife Research Center (NWRC) Utah Field Station located in Millville, Utah. This research facility maintains roughly 100 coyotes that are housed to aid research aimed at minimizing conflict between predators and people. Thus maintaining natural behavior of coyotes through humane care and enhanced enrichment are high priorities of the facility (*Schultz & Young, 2019*; *Shivik et al., 2009*). Coyotes are kept in enclosures that vary in size from 0.1 to 1.0 ha. Each pen generally houses a pair of coyotes (one male and one female), with breeding tightly controlled; none of the pairs we utilized in this study were pregnant. We used coyotes housed in three pen sizes: 0.1, 0.6 and 1.0 ha. We do not think pen size had any effect on our experiment because the study arena we established (see "Pen Setup" below) was the same in all the experimental enclosures and there was ample room in even the smallest pen for coyotes to avoid the experiment if they desired (*i.e.*, the experimental setup occupied <5% of the smallest pen).

This study included 16 pairs of coyotes that were not enrolled in other studies and had not been involved in previous research projects with similar treatments (*e.g.*, exposure to light deterrents). We conducted trials during March and April 2022. Throughout the trials, water was provided *ad libitum* and normal daily food rations consisted of ~640 g of a commercially prepared food (Fur Breeders Agricultural Cooperative, Logan, UT, USA) per coyote. This food was also used for the six experimental baits offered to each coyote pair, using ~40 g per bait. Normal feeding occurred during the day and the trials occurred at night so there was no interference of the trials on feeding patterns. The protocol was approved prior to the study by the USDA-NWRC IACUC under QA3401.

## Deterrent vehicle

To carry out our experiment, we built a customized deterrent vehicle by mounting a Foxlight (Foxlights Australia, PTY LTD; https://www.foxlightsaustralia.com.au/about-foxlights/) on a 1/10 scale remote controlled (RC) vehicle. The Foxlight utilizes multiple colored LEDs that fire randomly in a circular array with the goal of deterring predation (*Hall & Fleming, 2021*; *Naha et al., 2020*). The RC car was approximately 50 cm long and 15 cm tall. With the Foxlight mounted to it, the complete unit stood approximately 30 cm. The deterrent vehicle was controlled by an observer who could switch the Foxlight on and off *via* a remote switch and control the vehicle's speed and direction.

## Pen setup

We conducted the trials in test arenas established in a small portion of each coyote pen (Fig. 1). To establish each test arena, we secured four rebar stakes into the ground so that approximately one meter of each stake was above ground. These were our marker stakes that defined the perimeter of our area of defense. Stakes were placed in the corners that defined a square with ~4.25-m sides (Fig. 1). The base stake, which was located closest to the fence line, marked the base location from which the vehicle deterrent operated (*i.e.*, starting and resting position). Three baits were placed inside the protected zone, midway between the base stake and each of the radiating stakes (*i.e.*, inside the movement zone). Additionally, three outside baits (*i.e.*, outside the movement zone) were placed 1 m farther out from each of the three radiating stakes in the line formed by the base stake and each of the radiating stakes. In each pen, we mounted two motion activated trail cameras onto the perimeter fence to capture any activity within the test arena. Trials occurred from approximately 8 pm–2 am, depending on number of trials that needed to be conducted and duration of each trial. All trials started after twilight to ensure consistent and full effect of the Foxlight. It is possible that the natural activity cycle of coyotes (*Andelt & Gipson, 1979*) influenced their behavior and reaction to the treatments; however, we saw no evidence of this from our observations as coyotes were consistently motivated by the food rewards as part of the trials. To aid in recording data and controlling the proper movement of the deterrent vehicle, we also mounted a light to the enclosure fence above the entrance to illuminate the arena while performing the test trials. This light produced 500 lumens and was enough to dimly illuminate the test arena so coyotes could be observed taking

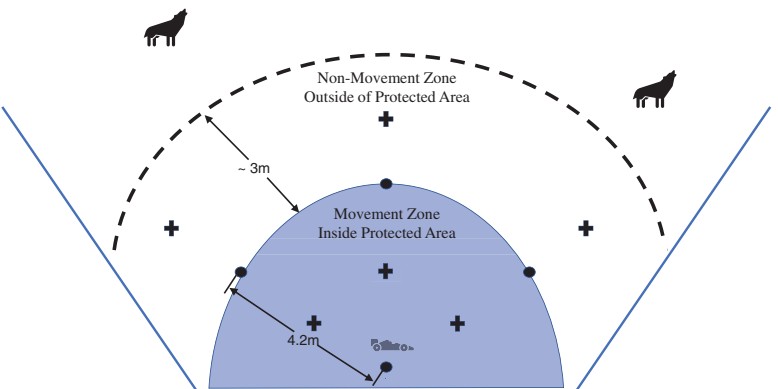

**Figure 1 The experimental arena established in each coyote pen.** The car represents the deterrent vehicle at the base station, the plus signs are bait piles, and the circles represent the marker stakes, with baits placed either inside or outside the movement zone. The movement zone represents the area where the vehicle was allowed to move (predetermined or adaptive) and the dashed line indicates the deterrent trigger line where we began adaptive movements if a coyote crossed this line. The RC vehicle remained/returned to the home station when not otherwise in use (predetermined route or adaptive pursuit).

baits. The light also likely affected the impacted of the Foxlight, but this impact was the same across all three movement treatments (described below).

## Experimental treatments

We established a treatment strategy with two levels of risk (baits inside and outside of the movement zone) and three levels of "movement". For the three movement treatments, the Foxlight was activated at the beginning of the trial and remained on for the duration. For the light only treatment, the vehicle remained stationary at the base position (Fig. 1) for the duration of each trial. For the predetermined movement trials, we drove the deterrent vehicle around the perimeter of the movement zone at regular three-minute intervals, with the deterrent vehicle returning to rest at the base position when it was not in motion. We chose 3 min per movement cycle because our goal was to have the deterrent vehicle in motion approximately 10–20% of the time and it took approximately 20–30 s to complete a loop around the perimeter. For the adaptive movement trials, we drove the deterrent vehicle toward a coyote that approached within approximately 5 m of the movement zone (*i.e.*, trigger line-dashed line in Fig. 1), taking care not to drive outside of the movement zone. If the coyote retreated, the vehicle was then directed back to the base stationary position. If the coyote moved to another part of the arena, we then followed the coyote with the deterrent vehicle, again taking care that it stayed within the movement zone. If the coyote retreated beyond the trigger line, the vehicle returned to the base station. Any uneaten baits were collected after the test trial was completed to prevent coyotes from obtaining baits outside of the test trial.

Each test enclosure had two coyotes and we did not restrict either coyote from participating in the trials. For each pen and each trial, we noted whether one or both coyotes attempted to retrieve baits, and this usually became apparent in the "baseline"

**Table 1 Description of trial details, including decision-making around the inclusion of pens, the collection of baseline data, and the collection of treatment data for the experimental trials.**

| Step | Duration | Description of step | # of pens completed |
|---|---|---|---|
| Arena acclimation | 7 Days | On day 1, place marker stakes and wildlife cameras in the pen to define the arena and allow coyotes to get used to these additions. *Behavioral criteria to pass to next stage: none.* | 16 |
| Bait acclimation | 2–3 Nights | At night, drive to within 5 m of pen gate, enter pen, place six baits in arena (three in the protected zone and three in the nonprotected zone), exit pen, immediately drive away. *Behavioral criteria to pass to next stage: none.* | 16 |
| Light/observer acclimation | 2 Nights | At night, drive to within 5 m of pen gate, enter pen, attach arena light to fence and turn it on, place six baits in arena, exit pen, sit in car and observe for 10–15 min. *Behavioral criteria to pass to next stage: at least one coyote in pen would come up and take at least one bait while observer was present.* | 13 |
| Baseline trial [Record data] | 1 Night | Same as above. Start timer as soon as last bait is placed and record the order and time until each bait is taken. Trial stops as soon as last bait is taken or 60 min has passed. | 13 |
| Light only trial [Record data] | 1 Night | Same as above except place the vehicle/deterrent in base position before placing baits and turn on Foxlight when last bait is placed. | 13 |
| Predetermined movement trial [Record data] | 1 Night | Same as above except drive vehicle/deterrent on fixed route (see "experimental treatments" for more detail) as soon as last bait is placed and Foxlight is turned on. Only half of filtered pens randomly selected for this trial. | 7 |
| Adaptive movement trial [Record data] | 1 Night | Same as above except drive vehicle/deterrent by adaptively reacting to coyotes (see "experimental treatments" for more detail). Only half of filtered pens randomly selected for this trial. | 6 |

phase (see Table 1). We ensured that as trials progressed, pens with either one or two coyotes interacting with the baits were balanced between the nonadaptive and adaptive treatments. Because sample sizes were so small, we did not include this as a covariate in the analysis. Instead, we relied on the fact that both the nonadaptive and adaptive treatments had a roughly equal number of pens with either one or two coyotes interacting with the baits and randomly assigned the last two treatments accordingly. The lead and second author carried out the randomization process and were aware of the group allocation throughout the experiment and the analysis.

## Trial sequence

We conducted trials during two periods (March and April, 2022); each period lasted approximately 3 weeks. The sequence and details of each trial are provided in more detail in Table 1. We used an observation vehicle to drive to each pen where up to three observers in the vehicle remained stationed for all aspects of the trial. The type of vehicle remained the same for the entire duration of the study. Only one person would exit the vehicle to set up a trial in a pen (see Table 1) and then would immediately return to the vehicle for the duration of the trial. Our study required the observation vehicle to be in close proximity (<5 m) to the test enclosures so we could record data and control the deterrent vehicle. Some coyotes were too "shy" (*Darrow & Shivik, 2009*; *Reale et al., 2007*) to approach the experimental area (*i.e.*, the animals within the pen would hide in their den boxes with observers present) and as a result we excluded three out of 16 coyote pairs during the "light/observer acclimation" period (see Table 1 for exclusion criteria). The remaining 13

**Table 2 AFT model ranks, based on Akaike Information Criteria adjusted for small sample sizes (AICc).** Results of the modeling effort to determine which model fits the data best. Movement indicates a treatment effect associated with the three treatments related to movement; bait position indicates a treatment effect associated with baits either inside or outside protected zones; the "+" indicates an additive effect; and the "*" indicates an interactive effect. K is the number of parameters; delta AICc is the difference in AIC score between the best model and the modle being compared; AICc Wt. is the proportion of the predictive power provided by each model; and LogLik is a measure of how each model fits the data.

| Model | K | AICc | Delta AICc | AICc Wt. | LogLik |
|---|---|---|---|---|---|
| (1) Movement * bait position | 9 | 1,127.25 | 0.00 | 0.92 | −554.22 |
| (2) Movement + bait position | 6 | 1,132.14 | 4.89 | 0.08 | −559.89 |
| (3) Movement | 5 | 1,209.27 | 82.02 | 0.00 | −599.50 |
| (4) Bait position | 3 | 1,292.58 | 165.33 | 0.00 | −643.24 |
| (5) Null | 2 | 1,333.71 | 206.46 | 0.00 | −664.83 |

coyote pairs were all subjected to the baseline data gathering and light only treatment, after which they were randomly allocated to either the predetermined movement or the adaptive movement treatment group. Because these last two treatments were of the most interest to us and we believed each treatment could potentially influence the outcome of the other, we randomly ascribed each coyote pair to one of these treatments instead of subjecting the same pairs to both treatments. We did not perform repeated trials on the coyotes (*i.e.*, animals were tested across treatments only once) because we believed there was a chance that coyotes could habituate to the deterrent and we did not want that to be a confounding factor in our experiment.

## Analysis

For all trials, we quantified the amount of time that each of the six baits survived. We used this duration data as our response variable in a time-to-event analytical framework, and right-censored data values from baits that remained uneaten after 60-min. We initially tried to use a Cox proportional hazards model to analyze the data but a Schoenfeld test for non-proportionality resulted in low *p*-values, indicating a violation of assumptions for this model. Thus, we fit Accelerated Failure Time (AFT) parametric models (*Wei, 1992*) using the 'survreg' function from the 'survival' package in R (*Therneau, 2021*). We tested for the best-fit probability distribution in our global model using visual assessment and AIC ranking (*Burnham, Anderson & Huyvaert, 2011*). The lognormal distribution had the best fit, based on its Akaike Information Criteria (AIC; model weight = 0.81), so we used this probability distribution for all competing AFT models. We fit five competing models (see Table 2) that included: a null model (no covariates), two single-covariate models (one with the movement treatment and the other with the risk treatment) and two models that included both movement and risk treatments (one using additive terms and the other using interaction terms). We used AICc (*Burnham & Anderson, 2004*) to rank each model and based our inference from the top model. We upheld a 0.05 significance threshold for our interpretation of results.

## RESULTS

We found strong support that bait survival was impacted by both the movement and risk treatments and that there was a significant interaction between these treatment types (Model 1 in Table 2, AICc Weight = 92%). This indicated that differences in bait survival were due to interactive differences between levels in the movement treatment and those in the risk treatment (Chi Square = 11.3, df = 3, $p$ = 0.010; Fig. 2; Table 3). Comparing treatment groups across the risk areas, survival of baits inside the protected zone was greater than bait survival outside the protected zone (Chi Square = 15.3, df = 1, $p$ < 0.001; Fig. 2; Table 3). Significant differences also varied among treatment types (Chi Square = 109.9, df = 3, $p$ < 0.001; Fig. 2; Table 3). Of note is that the light only treatment enhanced survival only inside the protected zone; outside baits for this treatment were estimated to survive the same amount of time as outside baits in the baseline group (Table 3). For baits inside the protected zone, a pattern of increasing survival is evident when comparing treatment groups in the following order: baseline—light only—predetermined movement—adaptive movement (Fig. 2; Table 3). Outside baits in the adaptive movement treatment were predicted to survive 80 min, compared to <3 min for any other treatment (Table 3). Survival curves of outside baits in the baseline, light only, and predetermined movement were predicted to be similar, but outside baits in the adaptive movement treatment group demonstrated a distinctly higher survival rate (Fig. 2). Twenty-one more baits inside the protected zone survived the entire trial duration than baits located outside the protected zone, with the majority of uneaten baits occurring in the adaptive movement treatment (Table 4).

## DISCUSSION

We found compelling evidence that creating wildlife deterrents that incorporate movement, particularly movement directed at and in response to animals (*i.e.*, adaptive movement), could provide superior aversive efficacy over devices that are stationary (Fig. 2; Table 3). Movement is important because it can influence all three concepts of deterrent effectiveness (*i.e.*, proximity, unpredictability, and adaptiveness). Our work addresses the effect of proximity at a local scale and our results show that proximity was an important factor when measuring the efficacy of a deterrent; baits further from the deterrent vehicle (*i.e.*, outside the protected zone) were almost always selected first and had lower survival probabilities (Fig. 2; Table 3). Field studies of deterrent efficacy also demonstrate that placement of deterrents in closer proximity to resources being protected is generally better than a deterrent away from the protected resources (*VerCauteren et al., 2020*). Having a deterrent capable of moving across the landscape is relevant because it could affect proximity dynamics by moving closer to resources when they are in immediate need of protection, enhancing its overall efficacy. Although, a deterrent moving closer to one object could also move further away from another in need of protection. This emphasizes the importance of spatially managing resources like livestock to maximize the effectiveness of a deterrent.

Movement of the deterrent vehicle also enhances unpredictability. Movement is already incorporated into deterrents like fladry that incorporates flags that wave erratically in the

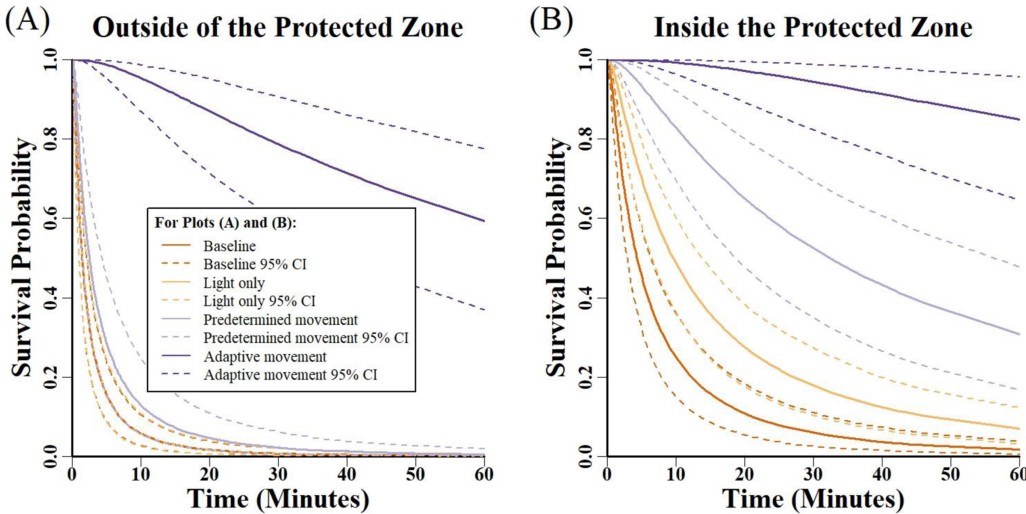

**Figure 2 Survivorship curves and corresponding 95% confidence intervals for baits made available to coyotes.** (A) baits outside the movement zone and (B) baits inside the movement zone. Both show survivorship for baits in the baseline trials and the three movement trials. A good indication of strong "significant" differences between treatments is seen where confidence intervals do not overlap for varying treatments. In (A) the curves for the baseline and light only treatment are almost completely congruent and overlap with the predetermined motion treatment.     

**Table 3 Predicted survival, lower 95% confidence interval (CI), and upper 95% CI of baits offered to coyotes during the study.** Results are from the top model (*i.e.*, "Movement * Bait Position" in Table 2) of the analysis to determine the impact of three movement treatments (light only, predetermined, and adaptive) across two levels of risk (inside protected zone and outside protected zone). Baseline refers to the pre-movement treatment data collected.

| Movement treatment | Protected zone | Predicted survival (Min) | Lower CI (Min) | Upper CI (Min) |
| --- | --- | --- | --- | --- |
| Baseline | Inside | 4.3 | 3.0 | 6.4 |
| Light only | Inside | 9.6 | 6.5 | 14.2 |
| Predetermined | Inside | 32.3 | 18.2 | 57.6 |
| Adaptive | Inside | 214.7 | 92.1 | 500.8 |
| Baseline | Outside | 1.4 | 1.0 | 2.1 |
| Light only | Outside | 1.4 | 1.0 | 2.0 |
| Predetermined | Outside | 2.5 | 1.5 | 4.2 |
| Adaptive | Outside | 80.3 | 41.3 | 156.2 |

wind (*Lance et al., 2010*; *Musiani et al., 2003*). Similarly, the "scary-man" has a vaguely-human effigy moving wildly when activated (*VerCauteren, Lavelle & Moyles, 2003*). Development of a deterrent vehicle extends the use of movement by creating a deterrent that moves across space, instead of only moving in place. In our study, all coyotes watched the vehicle deterrent during the trials, presumably learning about it and perhaps attempting to overcome fear. We noticed that during the predetermined movement trials that if the deterrent vehicle was moving away from the coyote while driving on a predetermined route, coyotes took advantage of these opportunities to obtain baits. It is

**Table 4 Description of the number of baits uneaten by coyotes within the 60-min experimental trials *vs.* the total number of baits made available (uneaten/made available).** Data from the trials with three movement treatments (light only, predetermined movement and adaptive movement) and two levels of risk (inside protected zone and outside protected zone). Baseline refers to the pre-movement treatment data collected.

| Protected zone | Movement treatments | | | |
|---|---|---|---|---|
| | Baseline | Light only | Predetermined | Adaptive |
| Inside | 0/39 | 5/39 | 12/21 | 15/18 |
| Outside | 0/39 | 0/39 | 0/21 | 11/18 |

likely that unpredictability would be enhanced by varying the amount of time (*e.g.*, in our trial this was 10–20% of the time) the deterrent vehicle is in motion, as well as variably switching the direction of movement while on a set course. Such changes would not add additional sophistication to the deterrent vehicle and would likely enhance its efficacy.

Our results indicate that movement can have a strong impact on the concept of adaptiveness, particularly if a deterrent can move in response to animals. We simulated this type of movement in the adaptive movement treatment and recorded exponentially greater protection for baits inside and outside the movement zone (Fig. 2; Table 3). Our observations indicated that the movement of the deterrent vehicle toward a coyote seemed to keep coyotes "on edge" or off-balance, which both increased the coyote's time spent overcoming its fear of the deterrent and decreased their time spent focusing on obtaining baits. The profound impact of adaptive movement noticed in this experiment helps justify further development of this type of vehicle. Critical for advancing this idea is the creation of a mobile deterrent that can both identify and react to animals. Progress in this regard is being made with agricultural animals (*King et al., 2023*; *Li et al., 2022*; *Yaxley, Joiner & Abbass, 2021*), but we are unaware of any autonomous deterrent systems that incorporates adaptive movement for wildlife management.

The idea of utilizing movement to enhance the three concepts of deterrents is not novel, as it is naturally incorporated into the use of livestock guarding dogs (*Gehring, VerCauteren & Landry, 2010*; *Saitone & Bruno, 2020*; *Smith et al., 2000a*). Guard dogs can move across a landscape, they are unpredictable, and can adaptively respond to animals on the landscape. Furthermore, guard dogs operate with multiple sensory systems (*i.e.*, auditory, sight, and olfactory) that enable the detection and a specific reaction to targeted species on the landscape. Given the effectiveness of guard dogs compared to many other kinds of deterrents, the exploration of alternatives might seem redundant. However, guard dogs are not always the most effective response to all predators; they can sometimes develop behaviors that threaten the resource they protecting, and they may become aggressive towards humans (*Green, Woodruff & Tueller, 1984*). Guard dogs also often require additional care and provisions from their human caretakers, and despite these investments, are still killed by large carnivores not infrequently (*Gehring, VerCauteren & Landry, 2010*). Moreover, the development of mobile deterrents with adaptive capabilities could enhance a wide variety of agricultural and human health and safety sectors where

dogs are not used (*e.g.*, crop protection, nuisance bird management at airports and golf courses, and fisheries protection). Thus, we believe efforts to improve wildlife deterrents should utilize guard dogs as models—not to replace them, but to add yet another tool when dogs are not contextually appropriate, cost-effective or successful.

What is novel is trying to incorporate the idea of adaptive movement into new technology used to manage wildlife. The development of deterrent vehicles capable of movement and adaptation is possible today, since the technology currently exists and is readily incorporated in the robotics field (*Ghobadpour et al., 2022*; *Maldonado et al., 2008*; *Roshanianfard et al., 2020*). Developing a vehicle that can employ predetermined movement, without adaptive movement, is an easier step because the base technology is fairly developed. Other challenges to creating a device like this include the development of a robust and field-worthy deterrent vehicle that can operate in all weather conditions and on varied terrains, can operate day or night, has sufficient battery power or local power source, and can be deployed consistently and reliably. Our results indicate that a device that employs predetermined movement could roughly double the effectiveness of a stationary deterrent. The development of an adaptive vehicle that can detect the presence of a predator and respond to its movements involves greater technological challenges. In addition to building a robust and field-worthy deterrent vehicle, technological developments would need to include image recognition, machine learning, and autonomous vehicle movement. Such technology is currently being incorporated into autonomous systems (*Ghobadpour et al., 2022*; *Maldonado et al., 2008*; *Roshanianfard et al., 2020*) and thus is possible, but the major hurdle relates to its cost. However, given the potential gains in efficacy that our results indicate, further development of such a vehicle could be economically justified.

While technological advances are an important component of resolving human-wildlife conflict issues, we believe they should also be incorporated into agricultural practices that enhance their efficacy (*Bruns, Waltert & Khorozyan, 2020*; *Muhly et al., 2010*). Our results support the idea that reducing predation risk on the landscape (*e.g.*, by spatially managing livestock) is a fundamental component of minimizing human-wildlife conflict and can enhance the utility of any deterrent. We suggest agriculturalists first consider if spatial management of livestock is possible and practical (*Lesilau et al., 2018*). If the answers are "yes", then incorporating wildlife deterrents into these systems becomes more useful and cost effective.

We acknowledge some potential caveats and limitations of this study and in our findings. First, we emphasize that this research was conducted with captive coyotes and extrapolating specific zero-sum outcomes from captive trials directly to field conditions or other contexts is unrealistic. Instead, our findings must be utilized only as a guide based on our comparison of treatments so that we might gain an understanding of their relative impact. It is also a critical consideration that our experiments "filtered out" shyer individuals (*Darrow & Shivik, 2009*) and subsequently incentivized the remaining coyotes to continue making attempts in the test arena to attain the baits. Our experimental design thus offers a worst case scenario for evaluating deterrent effectiveness, as it is possible that shyer individuals would have a disproportionately stronger reaction to the deterrent

systems. Given this context, it is also not surprising that the light only treatment provided limited protection and only for baits closest to the deterrent vehicle. Ultimately, as technological advancements are made, it is important that scientists perform rigorous field studies to obtain better insight into the true value of various deterrent systems for different species (*Khorozyan & Waltert, 2019*). In this experiment, we did not evaluate the process of habituation and how movement would influence it but this would be an important next step if a deterrent vehicle is developed. In particular, understanding how habituation is influenced with multiple coyotes involved is particularly relevant given that an adaptive deterrent may be reacting to one individual while another induvial is observing the interaction. Finally, a key component to deterrent effectiveness is the utilization of such devices in a preventative context, prior to wildlife learning that they may attain food rewards from agricultural resources (*Much et al., 2018*). This concept was not included in our experimental design and thus we have no inference as to how this may or may not influence the relative impact of each type of treatment.

## ACKNOWLEDGEMENTS

We thank the staff of the NWRC Utah Field Station for their assistance with this project.

### Funding

This work was supported by the U.S. Department of Agriculture. The funders had no role in study design, data collection and analysis, decision to publish, or preparation of the manuscript.

### Grant Disclosures

The following grant information was disclosed by the authors:
U.S. Department of Agriculture.

### Competing Interests

Cameron Krebs is employed by Krebs Livestock, and Anthony J. Giordano is employed by S.P.E.C.I.E.S.

### Author Contributions

- Stewart W. Breck conceived and designed the experiments, performed the experiments, analyzed the data, prepared figures and/or tables, authored or reviewed drafts of the article, and approved the final draft.
- Jeffrey T. Schultz conceived and designed the experiments, performed the experiments, analyzed the data, prepared figures and/or tables, authored or reviewed drafts of the article, and approved the final draft.
- David Prause conceived and designed the experiments, performed the experiments, authored or reviewed drafts of the article, and approved the final draft.
- Cameron Krebs conceived and designed the experiments, performed the experiments, authored or reviewed drafts of the article, and approved the final draft.

- Anthony J. Giordano conceived and designed the experiments, performed the experiments, authored or reviewed drafts of the article, and approved the final draft.
- Byron Boots conceived and designed the experiments, authored or reviewed drafts of the article, and approved the final draft.

### Animal Ethics

The following information was supplied relating to ethical approvals (*i.e.*, approving body and any reference numbers):

The USDA-National Wildlife Research Center IACUC approved the study (QA3401).

### Data Availability

The code and raw data are available in the Supplemental Files.

### Supplemental Information

Supplemental information for this article can be found online at http://dx.doi.org/10.7717/peerj.15491#supplemental-information.

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
