# Peer review of "Integrating robotics into wildlife conservation: testing improvements to predator deterrents through movement"

_PeerJ, doi:10.7717/peerj.15491_

## Round 0.1 · original submission · Major Revisions

Thank you for submitting this interesting study to PeerJ. I regret that I am unable to accept the manuscript for publication, at least in its present form. However, I am prepared to consider a new version that carefully takes into account the suggested edits. The reviewers liked many aspects of your study but also highlighted important parts that require revisions. These need to be addressed in detail in a new version. Such a revised manuscript is likely to be reviewed again and there is no guarantee of acceptance. When you revise the study, please prepare a detailed explanation of how you have dealt with all the reviewer comments as well as my own ones.

Please pay close attention to the Methods section of the manuscript, and provide further information regarding the ethics of data collection. See this PeerJ link for guidance: https://peerj.com/about/policies-and-procedures/#animal-research. “We require all authors to comply with the 'Animal Research: Reporting In Vivo Experiments' ARRIVE 2.0 guidelines, developed by NC3Rs. A completed full (21-point) ARRIVE 2.0 checklist must be submitted as a supplemental file with any submission that describes an interventional study on regulated animals.”
"Manuscripts describing studies of wildlife must show in their Methods section that they adhere to the ARROW guidelines."

The overall understanding of the manuscript would greatly benefit from more information on relevant coyote behaviours in the introduction, especially regarding foraging and hunting. Currently, the introduction has almost no information on the species being studied.

L180-181. Did you consider recording the behaviours with cameras and coding the information later, so that observers would not be close to the animals? The manuscript mentions that some animals were shy, so what impact did close human presence have on the results?

L208-210. Please revise this text to make it clearer. It would read better if you can make coyotes and their behaviours (or the baits) the subjects of the sentences, and expand the text if necessary. The current text is not clear.
L238. Revise “depredator”.

Overall, The Discussion requires substantial changes. L251-253, this text (and the reference) is only tangentially related to the main topic being studied. There are no other references in the first paragraph of the Discussion, which is very unusual for this part of a manuscript. Similarly, L272-313, there is little attempt to discuss the findings in light of the latest literature on this topic. See for example: Artificial eyespots on cattle reduce predation by large carnivores. 2020. https://www.nature.com/articles/s42003-020-01156-0#Abs1

L382. Information missing.

Figure 1. Delete: “A diagram depicting”.
Please provide an indication of the scale or size in Fig 1.

Reviewer 1 ·

Basic reporting

It is an interesting paper definitely addresses a need of the scientific community. This paper is clearly written and well organized. The introduction and background are reasonable given the premise of the paper. The Figures and tables are comprehensive and helpful. The raw data was supplied. There are several areas in the manuscript that deserve improvements. Relevant prior literature should be appropriately referenced in both method and discussion.

Experimental design

In general, the experimental design was excellent and clearly written. Some minor changes or explanations are needed.

Validity of the findings

The results are reasonable given the experiments.

Additional comments

Materials and methods
The approach is sound, well-reasoned and appropriate for the study's aims.

Line 119-125: Could you explain more information about captive coyotes in your study?

Line 123: What is the layout for each pen? Does each pen close to the other? When the researchers were researching, whether other pairs of coyotes could see the light from their own pen.

L125-127: “This was the maximum number of coyote pairs that were available for this study, as other pairs were either enrolled in other concurrent studies or had been subject to similar treatments in the past that we felt might impact this study.”…Do you mean the 16 pairs of animals have never done any experiment before? It is better to state clearly in the methods.

Line 128: What are the normal daily food rations? Is the food enough for the animals? Whether could this affect the results of this study?

Line 132: Could you explain where is the design of the vehicle idea from? From any previous study? Could you provide any related references? Whether the design idea from the electronic frightening device? The reference might be related as below:
Linhart, S.B., Dasch, G.J., Johnson, R.R. and Roberts, J.D., 1992. Electronic frightening devices for reducing coyote predation on domestic sheep: efficacy under range conditions and operational use. In Proceedings of the Vertebrate Pest Conference (Vol. 15, No. 15). 47, 386-392

VerCauteren, Kurt C.; Lavelle, Michael J.; and Moyles, Steve, "COYOTE-ACTIVATED FRIGHTENING DEVICES FOR REDUCING SHEEP PREDATION ON OPEN RANGE" (2003). USDA Wildlife Services - Staff Publications. 285,146-151

Line 155: “All trials occurred at night.”… Please state the time period was conducted in the methods. From the raw data file, whether the time conducted the experiment such as about 9 pm, 10 pm or 2 am could affect your results or not? Any reference to explain it?

Line 165: “we drove the deterrent vehicle around the perimeter of the movement zone every three minutes”…Could you explain why use every three minutes? Any previous study or reference could provide?

Line 180: How many observers are to be close to the pens to collect data?

Line 181-182 Some coyotes were too “shy” (i.e., the animals within the pen would hide in their den boxes with observers present). Could you provide the reference to define “shy”?

Results
L214-223: Is there any significant difference shown in Figure 2? Please explain the results clearly, both in the context and in Figure 2.

L224: Could you explain the reason for the study time “we capped the trials at 60 minutes”? Is there any reference provided?

Discussion
The study fails to address how the findings relate to previous research in this area in the discussion. The authors should rewrite their Discussion to reference the related literature.

Line 232-252: There is no reference. Could you add some references to support your ideas?

Line 252 and Line 285: Please check the reference format of PeerJ for the webpage.

Line 253: Please check the reference format of PeerJ for the book. Is the book name “Livestock Handling and Transport”? Please add the editor’s name and book name in the reference.

Line 272-313: There is no reference. Could you add some references to support your ideas?

·

Basic reporting

In the Introduction section lines 59- 65 you mention that coexistence with wildlife would benefit from the use of technology-based solutions with agricultural practices. It would strengthen your argument if you were able to provide some examples of other species/ fields of where this has been useful. For example, we know that UAV’s have been used to keep elephants away from human-dominated landscapes (Drones steer wandering elephants away from danger (newatlas.com)/ (Drone video of Chinese road-tripping elephants in slumber hits the spot (dronedj.com)) or monitor leopard movement (Forest dept uses thermal drones to trace leopard in Mysuru | Bengaluru - Hindustan Times). King et al., 2023 (Biologically inspired herding of animal groups by robots - King - Methods in Ecology and Evolution - Wiley Online Library) have recently published a paper on bio-herding of livestock which might offer parallels to the use of robots to influence animal movement and prevent conflict.
In lines 80-82 you mention “In developing more effective carnivore deterrents, at least three concepts are important: sensory diversity, adaptiveness, and mobility”. Could you provide more justification as to what makes these concepts important from the perspective of mitigation measures, specifically for sensory diversity and mobility? For example, Leveraging Motivations, Personality, and Sensory Cues for Vertebrate Pest Management - ScienceDirect. Similarly, Miller et al., 2016 in their review Effectiveness of contemporary techniques for reducing livestock depredations by large carnivores - Miller - 2016 - Wildlife Society Bulletin - Wiley Online Library found that light and sound were effective sensory cues in reducing carnivore depredation on livestock.

In lines 93-95 you mention ways in which management of livestock is used to reduce predation. Could you provide references for the examples you have alluded to.

The authors have systematically justified the use of the AFT model and the use of interactive terms to best describe the patterns seen. Complementary to/ instead of the raw data figure showing the difference between treatments and zones, it might be more interesting to see the survival curves for the model chosen. As the analysis which they have chosen looks at the time each bait has survived in the treatment groups. The authors have already run the code for this. This would allow the readers to see a visual representation of the results, when authors state that baits in the protected area survived xyz v/s baits in the non-protected area survived xyz. In addition, it would also help to have a table showing the results of the analysis, as now it is unclear how the p-values have been derived and the difference of each level of the model in comparison to each other. The authors briefly mention “Comparing treatment groups across the risk areas, survival of baits inside the protected zone was greater than bait survival outside the protected zone (p < 0.001). Significant differences also varied between treatment types (p < 0.001)”. It might be beneficial to see these results as a table as well.

The authors could create a table for the survival of the uneaten baits (lines 225-229) which would allow readers to make direct comparisons.

For table 1 the readers would benefit from knowing about the specific behaviours the authors were observing for the animal to successfully pass the stage. For example, in the stage of “Area acclimation”, exploratory behaviours such as sniffing, walking up to <1m away from the stake etc. This would give clarity into what allowed certain animals to pass the “acclimation” stage.
The structure of the article follows the recommended guidelines set by the journal. The figures and the tables titles explain the content well. However, the “Table 1” and “Table 2” in the text should be in italics as per the guidelines. Similar for the figure citation in text use “Fig” or “Figure 1” when starting a sentence with the citation.

There is a spelling error in the ‘Table 2’ legend “model”.

The supplementary raw data is accessible; and the R code is well described and transparent to the reader. Providing a key of the variables used for the analysis and what they represent in the raw data could benefit the reader.

Experimental design

In the materials and methods section could the authors provide some additional information on what baits were used? Were these baits different from the daily food rations that the coyotes were provided as that could have an influence over the motivation of the animals to procure the food baits.

The authors could mention the justification of why the animals were tested across treatments only once. For example, to prevent habituation from affecting the results (assuming which is why they were only tested once).

The authors state that there were two coyotes per experimental site. During the treatment phases did both the coyotes participate or was only one allowed to? Could this have had any potential impact on your results? Stating that in the manuscript would be important for the sake of reproducibility.

Validity of the findings

In the discussion section lines 232- 253 the authors have discussed important concepts of “predictability”, “proximity” and “intention” to deter “conflict” behaviours. The authors should provide additional references from the human-wildlife conflict literature which discuss these aspects. Or if there is insufficient literature then that would be an important gap which the authors would be filling in with their study. For example, the concept of “proximity” in the conflict literature has centred around the fact that agricultural lands/ communities closer to protected areas face the highest depredations by carnivores and herbivores Assessing Patterns of Human-Wildlife Conflicts and Compensation around a Central Indian Protected Area | PLOS ONE. Similarly regarding “predictability” the use of “lion lights” in Africa have shown degrees of success in warding off lions from cattle bomas Effectiveness of a LED flashlight technique in reducing livestock depredation by lions (Panthera leo) around Nairobi National Park, Kenya | PLOS ONE. Addressing these concepts in relation to currently used techniques could enhance the field of applied conservation further.

Additional comments

Overall, I found the paper to be well-written, easy to understand and appreciate the authors innovative use of technology to advance the knowledge in the field. While there is a need to use enhanced methods to mitigate negative human-wildlife interactions I would have liked to see more literature to justify the methods the authors chose; and the gaps in knowledge which they cover through their use of robotics. For example, the use of UAVs to mitigate human-wildlife conflict is gaining popularity, and the authors could mention the benefits of using their method (cost/animal behaviour/terrain) over the currently used techniques. I commend the authors honesty on the scope of their method and do agree that it could act as a complementary technique for existing traditional methods. As the authors have stated this method hasn’t been tested as a preventive measure for killing of livestock and future directions could include testing this in real-time conditions to test its efficacy.

---

## Round 0.2 · Minor Revisions

Thank you for carrying out comprehensive revisions to the manuscript. One of the original reviewers was not available, so the revised version was checked by only one of the original reviewers and two new ones. The reviewers and I are largely happy with the current manuscript, but nevertheless some further minor revisions are required. I would be very grateful if these new edits could be carried out and they will further enhance the quality of this study. When you revise the study, please prepare a detailed explanation of how you have dealt with all the reviewer comments.

·

Basic reporting

No comment

Experimental design

No comment

Validity of the findings

No comment

Additional comments

The authors have successfully incorporated the suggested edits. They have made significant improvements to the introduction and discussion sections of the manuscript. This provides a stronger argument for the methods they have employed and explaining the results they have obtained. They have provided justification for the use of the three concepts of proximity, unpredictability, and adaptiveness of the deterrents.

As a recommendation for the caveats of the study-the authors state that they did not restrict both the coyotes from participating in the trials. While I understand that there were no repeated trials it might be interesting to note in the discussion the potential implications of prolonged exposure to the adaptive movement. For example, when hunting in pairs/ packs could coyotes develop a habituation to the deterrents or use group hunting strategies? Possibly using this in conjunction with guard dogs might be useful as a double-deterrent method.

Reviewer 3 ·

Basic reporting

Thank you for the opportunity to review this work. I note that while I am a new reviewer to this manuscript, comprehensive work has been done to improve the manuscript following a constructive previous round of review and as such my comments are minimal.

The manuscript is clear and well-written, with sufficient background and references provided. I found it an enjoyable and interesting read. I have one minor comment which may further improve clarity. The words “adaptiveness”, “adaptability” and “adaptivity” are all used in the manuscript to refer to the same concept – I suggest using one of these terms consistently.

The colours of the lines plotted in Figure 2 do not exactly match the colour scheme of the legend, so I found this plot difficult to interpret. The other figure and the tables are clear and effectively convey the study results.

Experimental design

The experimental design is well thought-out and clearly explained. Information on study animal welfare is provided, including details of institutional ethical approval.

Validity of the findings

Raw data and code are included in the submission. Conclusions are well-stated and in accordance with the Results in accordance with PeerJ guidelines. Statistical methods used are robust and appropriate.

Additional comments

Line 134: Specify what RGB stands for.

·

Basic reporting

- The paper is generally clearly written. I’ve given a few specific suggestions for wording and identified places that could use clarification in the General Comments section.
- The background information in the Introduction is generally sufficient, although see the General Comments section for a couple of suggestions.
- The structure of the article is good and the figures and tables are useful and appropriate.

Experimental design

- The research questions, experimental design, methods and analyses were all very clearly explained.
- Data are available and code seems complete and well-commented.

Validity of the findings

- The findings seem valid, with conclusions based solidly on the results of the study.

Additional comments

General comments
- Line 94 – unclear whether “those species” refers to livestock, carnivores, both?
- Line 100 – “reduce the number of carnivores that kill livestock” sounds like the goal is to reduce carnivore population numbers, rather than reduce livestock loss to carnivores. I suggest changing to “reduce livestock losses” or “reduce livestock losses to predation by carnivores” or similar.
- Line 123 – This is the first mention of FoxLight in the main text. Would it be possible to include a reference or the name of a manufacturer so unfamiliar readers can learn more? Even just a “see below” would help orient the reader by indicating that this device will be discussed in detail later.
- Line 125 – Until this point the paragraph has dealt with wildlife generally, but here adaptability is mentioned as important specifically for carnivores. Should “carnivores” be “wildlife”, to make this more general?
- Line 132 – “Adding mobility” – are there no existing examples of mobile deterrents? If not, please state this explicitly so it’s clear that this is a major gap you’re working to address with the current manuscript. I also suggest making this sentence the first of this paragraph – point out that a mobile stimulus can check all three boxes (proximity, unpredictability, adaptiveness), and THEN say that recent advances in autonomous vehicles make this achievable.
- Line 141 – This paragraph feels out of order, as it feels like we are now “backing up” from deterrents to other possible approaches for mitigating livestock predation. I suggest moving this paragraph earlier - perhaps before or combined with paragraph 3.
- Line 151 – I’m a little confused by what is meant by “tech-herding”. According to the text in this paragraph is it a combination of a deterrent with spatial livestock management practices aimed at reducing predation (e.g. by the text, using night pens plus a guard dog would count as “tech-herding”), but perhaps you mean specifically to refer to approaches that incorporate novel electronic/robotic devices?
- Line 160-161 – Although I recognize you are using coyotes as a model example of broader concepts relevant to many wildlife species, it would be good to summarize the findings of previous studies of the effectiveness of deterrents against coyotes specifically, particularly if you can do so in relation to the concepts already discussed (motion, proximity, unpredictability, adaptiveness). This would help further define gaps in previous work that your study will fill.
- Line 225-227 – How bright was this light, and how did it affect the strength of the Foxlight as a stimulus?
- Throughout the manuscript (esp Results section), I would find it clearer if the “No Movement” treatment was instead called “FoxLight only”. This makes the difference between this treatment and the baseline treatment more explicit.
- In general, I found the methods and analysis sections to be very clear and easy to understand. The methods and statistical analyses seem appropriate.
- Figure 2 might be a bit clearer if the confidence intervals were shown as shaded intervals surrounding each survival curve, rather than dashed lines.
- Line 405 – In this paragraph, I suggest that the authors mention that their study did not address the potential of the coyotes to habituate to the various stimuli. Given that this is a major limitation of other deterrent devices, it seems worth mentioning that habituation could lead to a degradation of deterrent effectiveness over time.

---

## Round 0.3 · accepted · Accept

Thank you for carrying out the final revisions.